# Can Vision-Language Models Enable More Efficient Concept-Based Learning with Less Supervision for Interpretable Lung Nodule Diagnosis?

**Baoqiang Ma**[1] ⓘD                                              B.MA-2@UMCUTRECHT.NL
**Djennifer Madzia-Madzou**[1]                          D.K.MADZIA-MADZOU-3@UMCUTRECHT.NL
**Jin Ouyang**[1]                                                   J.OUYANG@UMCUTRECHT.NL
**Kenneth Gilhuijs**[1]                                        K.G.A.GILHUIJS@UMCUTRECHT.NL
[1] *Image Science Institute, University Medical Center Utrecht, the Netherlands.*

**Editors:** Under Review for MIDL 2026

## Abstract

Interpretability is necessary for the safe deployment of AI systems in clinical practice, especially in tasks such as the diagnosis of lung nodules. Concept bottleneck models (CBMs) provide a promising framework for interpretable predictions by linking decisions to clinically meaningful concepts. However, standard CBMs rely on extensive and time-consuming concept annotations. Recent methods aimed to fill this gap by leveraging vision-language models (VLMs) for few-shot or even label-free concept learning. However, it remains unclear whether prior knowledge within VLMs is sufficient for fine-grained nodule-level concept detection. In this work, we comprehensively investigate how much supervision is essential for reliable concept-based diagnosis and whether VLMs can improve efficiency. We compare black-box models, standard CBMs, few-shot VLM-based CBMs, and label-free CBMs on CT-based lung nodule diagnosis. The results show that few-shot VLM-based CBMs achieve improved concept detection (Balanced accuracy (Bacc): 0.78 vs. 0.76, F1 score: 0.76 vs. 0.72) and diagnostic performance (Bacc: 0.72 vs. 0.52, 0.74 vs. 0.36) compared to standard CBMs, and can even outperform black-box models in F1 score (0.74 vs. 0.66). In contrast, label-free CBMs produce unreliable and meaningless concept representations. These results suggest that VLMs can reduce supervision and improve interpretability and diagnostic performance, but are not yet sufficient for fully label-free concept-based learning.
**Keywords:** Concept learning, interpretability, lung nodule diagnosis, Vision-Language model.

## 1. Introduction

Reliable clinical application of AI radiology tools requires not only strong predictive performance but transparent decision-making. Concept-based approaches, including Concept Bottleneck Models (CBMs) (Koh et al., 2020), aim to extract clinically meaningful concepts from images and use them for prediction. These models have been explored in CT-based lung nodule analysis on the LIDC-IDRI dataset (Armato III et al., 2011), which includes annotations of interpretable concepts such as spiculation and texture. However, a primary limitation of CBMs is their reliance on expensive and time-consuming concept annotations from radiologists.

This burden could potentially be addressed by recent progress in vision-language models (VLMs). VLMs such as CLIP (Radford et al., 2021) enable few-shot or zero-shot (label-free)

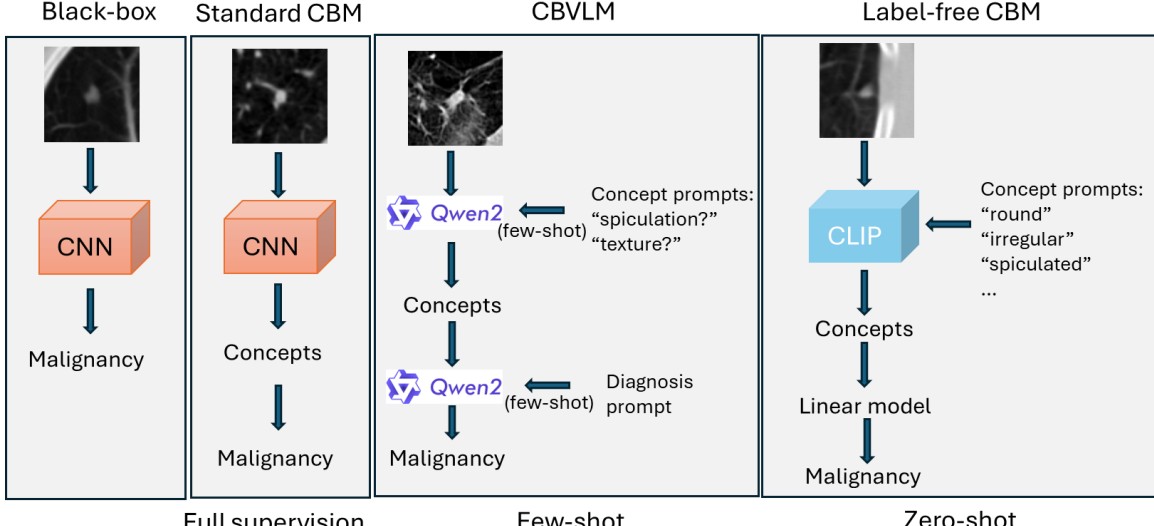

Figure 1: Overview of the comparison framework. We evaluate concept-based models under different supervision levels, including fully supervised CBMs, few-shot VLM-based CBMs (CBVLM), and zero-shot label-free CBMs.

concept learning by leveraging prior image–text understanding acquired from large-scale pretraining. Recent works have explored VLM-based concept models in few-shot setting (CBVLM) (Patrício et al., 2025), and have shown promising results across multiple medical imaging domains such as skin and chest X-ray imaging. In parallel, zero-shot approaches such as label-free CBMs (Oikarinen et al., 2023), typically built on CLIP, attempt to infer concepts without supervision and have shown strong performance in natural image domains. These approaches have also been extended to medical imaging tasks, such as skin and fundus image analysis (Chowdhury et al., 2024). However, these datasets involve well-defined and visually distinctive concepts, making them easier to detect than subtle, fine-grained nodule-related concepts in CT. It remains unclear whether such approaches generalize to more complex lung nodule characterization tasks.

This work aims to address this gap by investigating: (1) whether VLMs can reliably support concept learning for lung nodule analysis and (2) how much supervision is required to achieve effective and interpretable performance.

## 2. Materials and Methods

### 2.1. Data and Preprocessing

The public LIDC-IDRI dataset was used for lung nodule analysis. For each nodule, a 3D region of size $64 \times 64 \times 64$ mm$^3$ was cropped around the nodule center. The central axial slice was extracted and resized to $224 \times 224$ as model input. Malignancy labels and eight radiologist-annotated concepts (e.g., spiculation, texture) were binarized into $\{0, 1\}$. The

Table 1: Comparison of different models for concept detection and malignancy prediction.

| | Concept Detection | | Malignancy Prediction | |
|---|---|---|---|---|
| Model | Bacc | F1 | Bacc | F1 |
| Black-box | – | – | **0.83** | 0.66 |
| CBM (Full) | 0.76 | 0.72 | 0.52 | 0.36 |
| CBVLM (Few-shot) | **0.78** | **0.76** | 0.72 | **0.74** |
| Label-free CBM | – | – | 0.79 | 0.62 |

dataset was split into training, validation, and test sets, containing 1197, 211, and 213 nodules, respectively.

## 2.2. Models

We compare four settings, as shown in Figure 1 : (1) a black-box baseline and (2) a standard CBM, which first predicts concepts and then uses them for diagnosis, both implemented with a 2D ResNet50 backbone. (3) A few-shot CBVLM using Qwen2 for concept prediction via in-context examples and text prompts, followed by prompt-based nodule diagnosis. We use Qwen2 as the original authors showed general VLMs match or outperform medical-specific ones. (4) A zero-shot label-free CBM in which concept scores are extracted from a CLIP model via predefined prompts and fed into a linear classifier for malignancy prediction, following the official label-free CBM implementation.We also tried CT-CLIP (Hamamci et al., 2026) but neither model yielded meaningful concept scores (Figure 2).

## 3. Results

Table 1 shows the performance of all methods. CBVLM achieved better concept detection performance compared to the fully supervised CBM(Balance accuracy (Bacc): 0.78, F1: 0.76). For malignancy prediction, the black-box model achieved the highest balanced accuracy (0.83), while CBVLM obtains the best F1 score (0.74), indicating the better balance between precision and recall. The label-free CBM showed competitive balanced accuracy (0.79) but lower F1 score, suggesting its less stable predictions.

## 4. Discussion

Although Black-box model achieved strong predictive performance, but their lack of interpretability limits clinical use. Standard CBM achieved reasonable concept detection performance. However, its malignancy prediction is much worse than other models, likely due to misalignment between learned concepts and the target label.

CBVLM performs well in both concept detection and diagnosis. This is probably due to prior visual knowledge in VLMs and the use of few-shot guidance, which helps adapt the model to lung nodule analysis. In contrast, label-free CBMs fail to learn meaningful concepts in the experiments. This could be caused by no supervision and limited nodule-level concept knowledge in current VLMs.

Future work should explore better in-context example selection for VLMs and fine-tuning VLMs on nodule-level concept annotations to improve concept prediction quality.

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

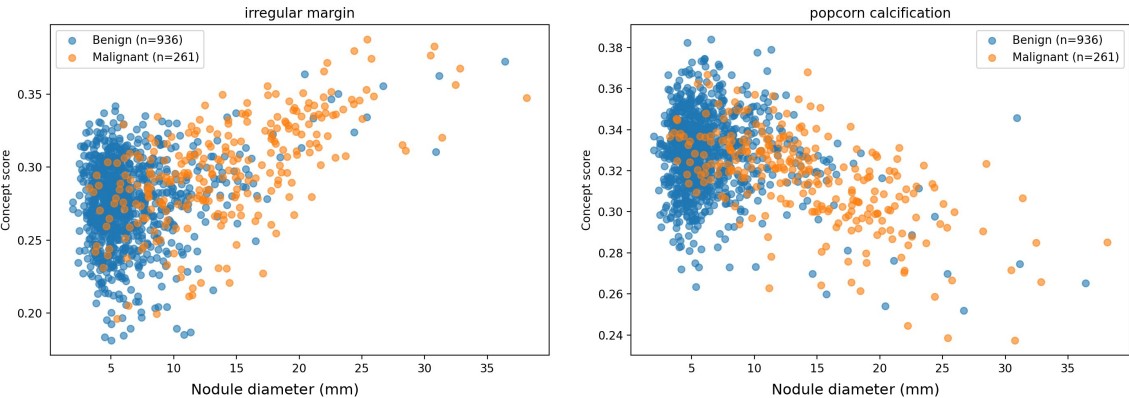

Figure 2: CT-CLIP concept scores for irregular margin and popcorn calcification on the training set, showing poor separation between benign and malignant nodules.

## 5. Appendix

Figure 2 shows the concept scores extracted by CT-CLIP on the training set for two example concepts: irregular margin and popcorn calcification. The extracted scores struggle to distinguish malignant from benign nodules, and this limitation becomes more pronounced for nodules with smaller diameters.

