# OpenReview forum: "Can Vision-Language Models Enable More Efficient Concept-Based Learning with Less Supervision for Interpretable Lung Nodule Diagnosis?"
_MIDL.io/2026/Short_Papers — MIDL 2026 - Short Papers Poster_

### Official Review · Reviewer_tcJP · 2026-04-29
**Review: Can Vision-Language Models Enable More Efficient Concept-Based Learning with Less Supervision for Interpretable Lung Nodule Diagnosis?**

**Rating:** 2
**Confidence:** 4

**Review:**

This paper is clearly written. In terms of quality, it is decent, with several small writing issues. It also lacks scientific depth in terms of discussion of current work or references to prior work. It compares four different models, but for each of these models, crucial implementation details are missing, while other parts could easily be condensed. The paper suggests that these are based on previous work but never actually mentions the previous work.

Judging from the current status of the paper, it sounds like it used three different Concept Bottleneck Models (CBM) that were previously proposed and compared them for the task of lung nodule analysis. The paper mentions that this is harder and different from what these models have been evaluated on previously but does not provide any evidence or details about it.

The detailed pros and cons of the paper are listed below.

The significance of this work is high, given that interpretability of current models is often ignored and very low but could help increase trust and overall understanding.

**Summary:**

This paper compares four methods for malignancy prediction in terms of interpretability using concepts. It compares how well the prediction of lung-nodule malignancy performs when using interpretable concepts rather than black-box models.

Comparing the methods shows that using a black-box model outperforms all three concept-based methods in terms of bAcc but not F1. In terms of concept detection, CBVLM retains interpretability, slightly outperforming full concept bottleneck models.

**Strengths:**

- The paper is well written and easy to follow.

- The comparison between four methods on malignancy prediction in terms of if and how well they can predict concepts is interesting.

- Leveraging large pretrained VLMs for concept-guided prediction is a good idea with great potential impact.

**Weaknesses:**

- The models used for CBVLM and label-free CBM (Qwen and CLIP) are not tailored toward medical reasoning. I believe LLMs/VLMs that were finetuned on medical data should be used, or at least compared to, especially if interpretability is a focus.

- Methodologically, it is hard to follow what the paper is actually doing. For example, how exactly is CLIP integrated for Label-free CBM? If this is done just like Chowdhury et al. (2024), then this should be properly mentioned.

- There are no concept-level metrics for label-free CBM. At that point, this is no longer a concept-based method.

- The conclusion section feels unpolished and lacks motivation. What does it mean to "combine few-shot learning into VLMs"? Also, what part of the paper led to the conclusion that VLMs should be trained on "nodule-level concepts"?

**Justification Of Rating:**

The paper presents a study on comparing different approaches for using concept bottleneck models for lung nodule malignancy prediction. While the problem itself is important, the approaches chosen for CBM are not tailored toward medical evaluation. Currently, the paper only compares these four known approaches on a new dataset with little information about the specific importance and difficulty of CBM in this setting. Consequently, I cannot recommend acceptance at the current stage.

---

### Decision · Program_Chairs · 2026-05-08

Accept (Poster)